# Green Reflector with Predicted Chromatic Coordinates

**DOI:** 10.3390/ma16062316

**Published:** 2023-03-14

**Authors:** Xin Tong, Zhuo Yang, Jiali Zhang, Wenbing Li, Bo Liu, Chang Chen

**Affiliations:** 1School of Microelectronics, Shanghai University, Shanghai 201800, China; 2Shanghai Industrial Technology Research Institute, Shanghai 201800, China; 3State Key Laboratory of Transducer Technology, Shanghai Institute of Microsystem and Information Technology, Chinese Academy of Sciences, Shanghai 200050, China; 4Shanghai Academy of Experimental Medicine, Shanghai 200052, China

**Keywords:** green reflector, inductively coupled plasma chemical vapor deposition, transfer matrix method, finite-difference time-domain, chromatic coordinates

## Abstract

The color reflector with multiple-layer thin film scheme has attracted much attention because of the potential for massive production by wafer-scale deposition and the possibility to integrate with photonics (semiconductor) devices. Here, an angle-insensitive green reflector with a simple multilayer dielectric thin film structure was reported, with predicted chromatic coordinates based on CIE 1931 standard. The SiN/SiO_2_ multilayer thin film stack, including a special silicon-rich nitride material with ultrahigh refractive index, was grown alternatively by an inductively coupled plasma chemical vapor deposition (ICPCVD) system at a low stage temperature of 80 °C. The green reflector showed a maximum reflectivity of 73% around 561 nm with a full width at half maximum (FWHM) of 87 nm in the visible wavelength range, which contributed significantly to its color appearance. The measurement by an angle-resolved spectrometer under the illumination of p/s-polarized light wave with a variable angle of incidence indicated that the reflectance spectrum blue-shifted slightly with the increasing of incident angle such that the green color could be kept.

## 1. Introduction

Color filters, including chemical filters and optical filters, were used to select light waves with specific wavelengths in the visible range, which led to wide applications in the field of display [1,2], imaging [3,4,5], color printing [6,7], and color decoration [8,9]. With advances of the high reliability under UV radiation and high-temperature environment, optical filters based on multiple-layer thin film, Fabry–Perot resonant cavity [10,11], photonic crystals [12,13,14], and gratings [15,16] are more popular in practical applications than the organic-dye-based chemical filters. Up to now, extensive research focused on various optical filters were reported, with some referring to the methods for improving the angle-insensitive property [17,18,19]. A subwavelength grating-based reflective filter fabricated by etching two-dimensional Au arrays on silicon-on-quartz (SOQ) substrate with the help of electron beam lithography (EBL) was presented [1]. It was reported to possess an angular tolerance of 45 degrees with the incidence of unpolarized light waves. Based on the classical structure of the Fabry–Perot resonant cavity, an angle-insensitive color filter combined with a sandwich-layer scheme of Ag–SiOx–Ag was proposed by Mao et al. [20]. It was reported that the filtered color could be tuned by modifying the refractive index of the middle layer SiOx in the cavity formed by the silver mirrors, by controlling the flow rate of O_2_ in the reactive magnetron sputtering chamber during materials growth. Further measurement results presented that high transmission was maintained without significant variation while changing the angle of incidence among a wide range between 0–60 degrees in the visible light range. With the advances of antioxidation in storage and low loss caused by avoiding metal absorption, all dielectric thin film optical color filters made by staggering deposited SiO_2_ layer and an a-Si:H layer on the glass substrate were reported to maintain good hue saturation among red light regions with an angle of incidence from 0 to 80 degrees [21].

Similar to color filter stacks on the transparent substrate, the multiple thin film layers can be deposited on silicon substrate as a reflector, to select the wavelengths and color of visible light waves to reflect back rather than transmit through the substrate. In this work, a simple multiple-layer stack combined of very-high-refractive-index, silicon-rich, highly nonstoichiometric amorphous silicon nitride (SiN for abbreviation) and low-refractive-index SiO_2_ was grown by an inductively coupled plasma chemical vapor deposition (ICPCVD) system continuously on an 8-inch silicon wafer at 80 °C, as an angle-insensitive reflector to reflect the green light wave centered at 561 nm. The result showed that the reflectance spectra blue-shifted with the increasing of incident angle for both the s and p-polarized light, and thus led to the tiny change of color coordinates in the CIE 1931 color space. The morphology characterization together with the calculated reflectance spectra based on it, and the related measured reflectance spectra, revealed that the deviation of practical thickness for each layer from the corresponding target thickness resulted in the shift of reflectance spectra and related color coordinates. Therefore, it provided useful information for further growth of the alternative SiN/SiO_2_ multiple thin film layer stack to link the design target and deposition process, for more complex layer structure.

## 2. Methods and Methodology

### 2.1. Basic Theory

#### 2.1.1. Transfer Matrix Method (TMM)

According to the transfer matrix method [22], the reflectance of an assembly of thin films was calculated by the optical admittance, obtained from the product of the individual matrix that represented the optical property in sequence. The complex refractive index, the physical thickness of each layer, and the angle of incidence for each layer were contained in the phase thickness. The reflectance and characteristic matrix of the assembly can be expressed as:(1)$$R=\left(\frac{\eta_{0}B-C}{\eta_{0}B+C}\right){\left(\frac{\eta_{0}B-C}{\eta_{0}B+C}\right)}^{*}$$
(2)$$\;\left[\begin{array}{c} B\\ C \end{array}\right]=\left\{\overset{q}{\underset{r=1}{\prod }}\left[\begin{array}{cc} cos\delta_{r} & \frac{i}{\eta_{r}}sin\delta_{r}\\ i\eta_{r}sin\delta_{r} & cos\delta_{r} \end{array}\right]\right\}\left[\begin{array}{c} 1\\ \eta_{m} \end{array}\right]$$
(3)$$\delta_{r}=\frac{2\pi }{\lambda }N_{r}d_{r}cos\theta_{r}$$
where $\delta_{r}$ represents the phase thickness of the film, $N_{r}$ represents the complex refractive index of the material, $d_{r}$ represents the optical thickness of the film, $\eta_{0}$ represents the admittance of the incident medium, $\eta_{r}$ represents the admittances of the film, and $\eta_{m}$ represents the substrate admittance.

The calculation for the reflectivity, transmission, and absorption of a multiple-layer stack under the illumination of s/p-polarized plane waves with an angle of incidence was distinguished by the admittance values for each layer. The p/s-polarized wave could be simply distinguished by the optical admittance of each layer with a tilted incident angle, which can be expressed as follows [22]:(4)$$\eta_{r}\{\begin{array}{c} \;N_{r}/cos\theta_{r}\;\;\;\left(p\;polarization\right)\\ N_{r}cos\theta_{r}\;\;\;\;\;\left(s\;polarization\right) \end{array}$$

#### 2.1.2. 2D Finite-Difference Time-Domain (FDTD) Algorithm

The FDTD algorithm is a method used to discretize the Ampere’s equation and Faraday’s equation, both in space and time domains, such that the time-resolved space dependent electric field and magnetic field are calculated at each Yee grid [23]. A Fourier transformation from the time domain covered the short period from the moment that the source disappeared to the moment that almost all the power in the simulation region was extinct, to the frequency domain that, among the wavelength range for the source, obtained the wavelength-dependent field at each Yee grid [24]. The update of the E/H field in finite space and time domain was connected by the light velocity at corresponding media such that the light waves propagated through, and the Courant stability condition ensured the stability of numerical calculation at each Yee grid for materials with different refractive index and dispersion relationship in the simulation region [23]. The multiple-layer thin film stack designed as a green reflector was composed of seven layers of ultrahigh-refractive-index SiN and low-refractive-index SiO_2_ alternatively stacking on a silicon substrate (Air/SiO_2_(118 nm)/SiN(280 nm)/SiO_2_(118 nm)/SiN(140 nm)/SiO_2_(118 nm)/SiN(70 nm)/SiO_2_(280 nm)/Sub). In a 2D Cartesian coordinate system, the y = 0 point was set at the interface of the bottom SiO_2_ matching layer and the silicon substrate. The perfectly matched layer (PML) [25] boundary was placed at the position with a distance about one wavelength from the top SiO_2_ layer to absorb the scattered light waves leaving the simulation object. With an assumption that the thickness of the silicon substrate was infinite, another PML boundary was placed in the substrate with a distance from the interface between the bottom SiO_2_ layer and silicon substrate (y = 0) about one wavelength, to absorb the transmitted light waves from multiple-layer stack to the substrate. On the x-axis, a width about several wavelengths, including the region between two PML boundaries, was selected as the simulation object, to represent the wafer-scale thin film stack on the silicon substrate. Two periodic boundaries were placed at the edge of the simulation object vertical to the x-axis. The simulation object was illuminated by a plane wave source with a broad wavelength range. In case the wave vector was vertical to the x-axis, the s-polarization and p-polarization waves were the same for the incident surface without any pattern. With an angle of incidence, the p/s-polarized source was used for the simulation, respectively. In addition, the broadband fixed angle source technique (BFAST) was introduced for the plane wave source, to ensure the wavelength was not dependent on the incident angle in the calculation [26]. The reflectivity of the simulation object was computed by the ratio of the power reflected over the power injected to the simulation region [23].

#### 2.1.3. Colorimetry

The human vision is sensitive to visible light with three different colors that cover 564 nm to 580 nm (orange–red), 534 nm to 545 nm (green), and 420 nm to 440 nm (blue) [27], the corresponding spectra of which are defined as the three primary colors that can compose all different colors in the world. The expressions for evaluation of the tristimulus values were followed as per [28]:(5)$$X=K\int_{380}^{780}S\left(\lambda \right)\rho \left(\lambda \right)\bar{x}\left(\lambda \right)d\left(\lambda \right)$$
(6)$$Y=K\int_{380}^{780}S\left(\lambda \right)\rho \left(\lambda \right)\bar{y}\left(\lambda \right)d\left(\lambda \right)$$
(7)$$Z=K\int_{380}^{780}S\left(\lambda \right)\rho \left(\lambda \right)\bar{z}\left(\lambda \right)d\left(\lambda \right)$$
where K is the adjustment factor, S(λ) is the ability distribution of the light source, ρ(λ) is the reflection performance of the object surface, and $\bar{x}\left(\lambda \right)$, $\bar{y}\left(\lambda \right)$, $\bar{z}\left(\lambda \right)$ are the characteristic parameters of the human color vision system.

According to the 1931 CIE standard [29], the chromatic coordinates were obtained from the tristimulus values, and the position in the 1931 CIE chromaticity diagram that the color of corresponding object could be recognized. The conversion formula is
(8)Y=Y
(9)$$x=\frac{X}{X+Y+Z}$$
(10)$$y=\frac{Y}{X+Y+Z}$$
where *Y* is the brightness factor, indicating the brightness of the color.

Thus, the color chromaticity coordinates can be calculated from the reflectance spectrum of an object.

### 2.2. Experiment

The Leuven ICPCVD (Leuven Instruments, Xuzhou, China), with the advance of an innovative configuration, consisted of two RF sources; the precursors can be mixed together and ionized by the source RF at first and can then be accelerated to the substrate by the bias RF sequentially. It makes the chemical vapor deposition happen at a substrate with a very low temperature, such as 80 °C [30].

Compared with the inherent high absorption of a-Si in the visible range, which makes it difficult to process, SiN has a smaller extinction coefficient in the visible range, and the refractive index and extinction coefficient of SiN can be adjusted by controlling the process parameters during ICPCVD deposition. The SiN thin film was grown by the chemical vapor deposition of ions species from SiH_4_ and N_2_, with a flow rate ratio that consisted of more SiH_4_ than necessary in the normal stoichiometry (SiH_4_:N_2_ = 55 sccm:30 sccm), to achieve a higher refractive index. Meanwhile, the SiO_2_ layer was deposited by the mixed plasma of SiH_4_ and N_2_O with a ratio that followed the stoichiometry (SiH_4_:N_2_O = 20 sccm:40 sccm). Besides the precursors involved in the reaction, inert gas Ar with a flow rate of 200 sccm was used to improve the ionization of reactants and to stabilize the deposition process. The thicknesses of the SiO_2_ layers were 118 nm, SiO_2_ films were deposited in two steps during the ICPCVD process, the deposition time of the first stage was 10 s, and the second stage was 85 s. The thicknesses of SiN layers were 70 nm, 140 nm, and 280 nm, and the deposition times were 33 s, 67 s, and 133 s, respectively. The recipe for deposition of SiN and SiO_2_ materials is displayed in Table 1, respectively, under specific source power and bias power. A single layer of the SiN and SiO_2_ thin film was deposited on an 8-inch silicon wafer with determined refractive index, respectively, to obtain the n/k values and deposition rate of thin film with the assistance of an ellipsometer. Therefore, the reflectance spectrum was calculated by the method elaborated above, based on the measured refractive index and dispersion relationship, to design multiple-layer stacks with specific thicknesses for each layer. In addition, the process of film deposition using ICPCVD can be roughly divided into several steps: pump, stable, deposition, purge, and pump, in which the growth parameters strongly relating to the optical constant of SiO_2_ thin film and SiN thin film are presented in Table 1.

The designed green reflector based on a multiple-layer SiN/SiO_2_ thin film stack on the silicon substrate was grown continuously in the ICPCVD chamber with a corresponding target thickness, following the recipe illustrated above for single-layer deposition.

The refractive indices of both the SiN and the SiO_2_ thin films on the silicon substrate were measured and analyzed by a Muller spectra ellipsometer (ME-SA-200, Wuhan Eoptics Corporation, Wuhan, China). The reflectance spectra of the wafer-scale green reflector were tested by a reflectance tester (Wuhan Eoptics Corporation, Wuhan, China) with the incidence of a light source vertically. For the detailed study of the green reflector under the illumination of p/s light with various incident angles and the layer structure of the multiple thin films stack, a small piece with a size of about 2 cm * 2 cm was cut from the whole wafer for inspection with the macro-angle-resolved spectrometer (Shanghai Ideaoptics Corporation, Shanghai, China) and for morphology characterization of the cross-section using a scanning electron microscope (SEM) (Hitachi, Tokyo, Japan).

## 3. Results and Discussion

The refractive indices and extinction coefficients of Si substrate, SiN thin film, and SiO_2_ thin film covered a broad wavelength range from 300 nm to 1000 nm, and are shown in Figure 1. The results of the reflectance spectra by measurement and calculation, the SEM image of the cross-section of the multiple-layer stack on the green reflector, and the chromatic coordinates calculated from reflectance spectra are displayed in Figure 2, Figure 3 and Figure 4, respectively.

With the refractive index of the silicon substrate known in advance, the refractive indices and extinction coefficients of SiN and SiO_2_ films were measured by a Muller spectra ellipsometer and fitted by the Cody–Lorentz model [31] for monolayer SiN thin film and the Cauchy model [32] for monolayer SiO_2_ thin film, respectively. With more silicon element in the SiN material than necessary in normal stoichiometry, the silicon-rich silicon nitride thin film showed a very high refractive index; a silicon-like dispersion relationship covered the violet and near-infrared region with a maximum value at 354 nm, and presented high absorption in the violet to green wavelength region.

As presented in Figure 2a, the reflectance spectra covered a wavelength range from 400 nm to 800 nm; four different test points selected randomly on an 8-inch wafer (P1, P2, P3, P4), were tested, respectively, under the incidence of a broad-wavelength light source vertical to the surface. It was evident that the maximum reflectivity appeared in the spectra at a wavelength about 561 nm, with a large FWHM value of 87 nm and a high reflectivity value of about 73%. The reflectance spectrum based on the designed multiple-layer scheme for the green reflector is shown in Figure 2a, indicating that there was a shift of the peak wavelength as large as 21 nm (at 582 nm). Furthermore, in order to investigate the mismatch between practical thickness and the designed thickness of each layer, a small piece with a size of about 2 cm * 2 cm was cut from the margin of the whole wafer, for morphology characterization with SEM. The multiple-layer structure is presented in the SEM image, with symbols describing the materials and related thickness values, shown in Figure 2b. For a comparison study, the practical thickness values and designed thickness values for each layer are shown in Table 2. In addition, the reflectance spectra with actual film thickness of each layer obtained from the SEM image were calculated by the TMM method. The results indicated that a maximum reflectivity about 79% occurred at 563 nm, which matched well with the test spectra. The slight deviation between the measured spectra of the four points on the reflector was mainly caused by the inhomogeneity of the film deposition. The deviation of the reflectance spectra of the practical measurement data from the simulation data of the designed structure was due to the deviation of thin film thickness for each layer during material growth. The small variation between the reflectance spectra of simulation with actual film thickness and the measured spectra is possibly attributed to measurement errors and scattering losses due to the unevenness of the film surface.

Based on the tested refractive indices of SiN and SiO_2_ that covered a broad wavelength range from 300 nm to 1000 nm, the FDTD method was used to simulate the reflectance of the green reflector under the illumination of p/s-polarized plane wave, respectively, for different incident angles. As presented in Figure 3c,d, the reflectance spectra of the designed seven-layer film stack were insensitive to the angle of incidence in the visible range. The reflectance spectra covered the wavelength range from 400 nm to 800 nm for a piece cut from the margin of an 8-inch wafer, obtained by the macro-angle-resolved spectrometer under the illumination of p/s-polarized source (Figure 3a,b). Furthermore, the simulated reflectance spectra with actual film thickness are shown in Figure 3e,f, revealing the spectra with actual film thickness corresponding to the tested reflectance spectra under p/s-polarization at different angles. Compared with the data from Figure 3c,d, the reflectance spectra in Figure 3e,f are much closer to the result obtained from practical measurement in Figure 3a,b.

Though the central wavelength and peak reflectance changed slightly with the increase of incident angle, as displayed in Table 3, it still can be seen as a kind of angle-insensitive reflector [8]. It was worth noting that with the increase of incident angle, the reflectance spectra gradually blue-shifted, and the reflectance peak gradually decreased. Furthermore, under the illuminance of p-polarized light, the FWHM of the reflector’s reflectance spectra gradually narrowed with the increase of the incident angle, while the FWHM of s-polarized light did not change significantly. The discrepancy between the experimental and simulated reflectance spectra was probably caused by the fast growth rate of each layer and the nonuniformity of the thin film on the whole wafer. Therefore, it was necessary to control the thickness of each layer exactly and the thickness uniformity of each layer, while depositing multiple-layer thin film stacks alternatively. It provided us with useful insights for future thin films deposition.

Due to its simple structure and excellent characteristics of high brightness and color saturation, the proposed seven-layer green reflector is highly attractive for potential applications covering imaging, color decoration, and color printing. Here, the final 8-inch green reflector with good color purity in this work is displayed in Figure 4a. In actuality, the photograph was taken with a titled angle view by a camera, and it seems that the color of the reflector almost remained the same from different angles through the naked eye, which indicates its angle-insensitivity property.

In order to evaluate the chromaticity of the green reflector, the chromaticity coordinates of the experimental and simulated reflectance spectra at different incident angles were plotted on the standard CIE 1931 color space, as shown in Figure 4b. It was obvious that the color coordinate values at different incident angles moved toward the green region in the upper left of the CIE 1931 chromaticity diagram as the angle increased. Even though the reflectance peak gradually decreased as the incident angle increased for both the p- and s-polarized light, the high color saturation and brightness of the green reflector were progressively enhanced due to the blue-shift of the reflection peak, and the narrowing of the FWHM gradually.

## 4. Conclusions

In summary, SiN and SiO_2_ films were prepared by ICPCVD at a low deposition temperature of 80 °C, and SiN films with a high refractive index were also prepared by adjusting the process conditions. The n/k values of SiN and SiO_2_ were obtained by the ellipsometer measurement and analysis, and then the data were used in the design and simulation of the green reflector. The 8-inch wafer-scale green reflector reported in this work has angle insensitivity up to 55 degrees, peak reflectivity exceeding 40%, and fine color saturation. The color of the green reflector was predicted by FDTD calculation, which matched well with the color coordinates obtained from the tested reflectance spectra. The difference between the reflectance spectra from the measurement and from the FDTD simulation was due to the deviation of thickness for each layer from the designed values. Therefore, it was critical to control the thickness of each layer in the stack exactly and ensure the uniformity of thickness well during thin film deposition. The green reflector deposited by the ICPCVD method at low temperatures in this work provided useful insights for future design and deposition of multiple thin film layer stacks, and showed potential application in CMOS image sensors, display, color decoration, and low-temperature polymer devices.

## Figures and Tables

**Figure 1 materials-16-02316-f001:**
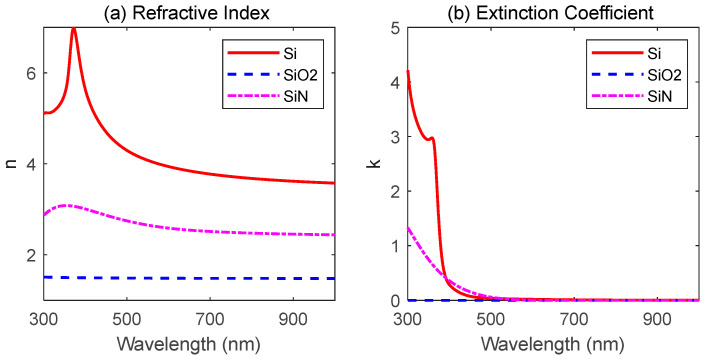
Dispersion relationship of Si substrate, SiO_2_ thin film, and SiN thin film used for simulation. (**a**) Refractive index and (**b**) extinction coefficient of each material.

**Figure 2 materials-16-02316-f002:**
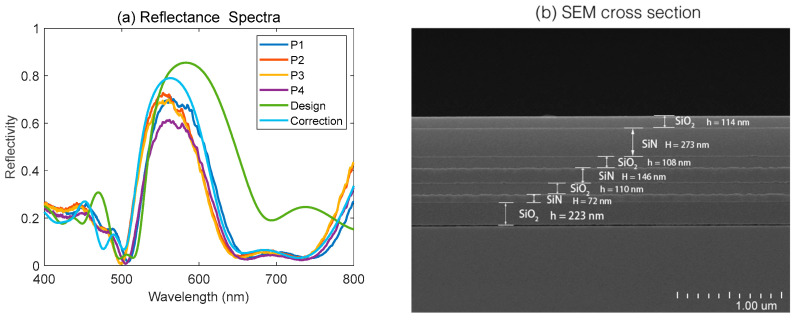
(**a**) The reflectance spectra of four randomly sampled points on the fabricated reflector, the simulated reflectance spectrum of the designed structure, and the reflectance spectrum by simulation with actual film thickness obtained from SEM image. (**b**) SEM image of the cross-section of the fabricated reflector.

**Figure 3 materials-16-02316-f003:**
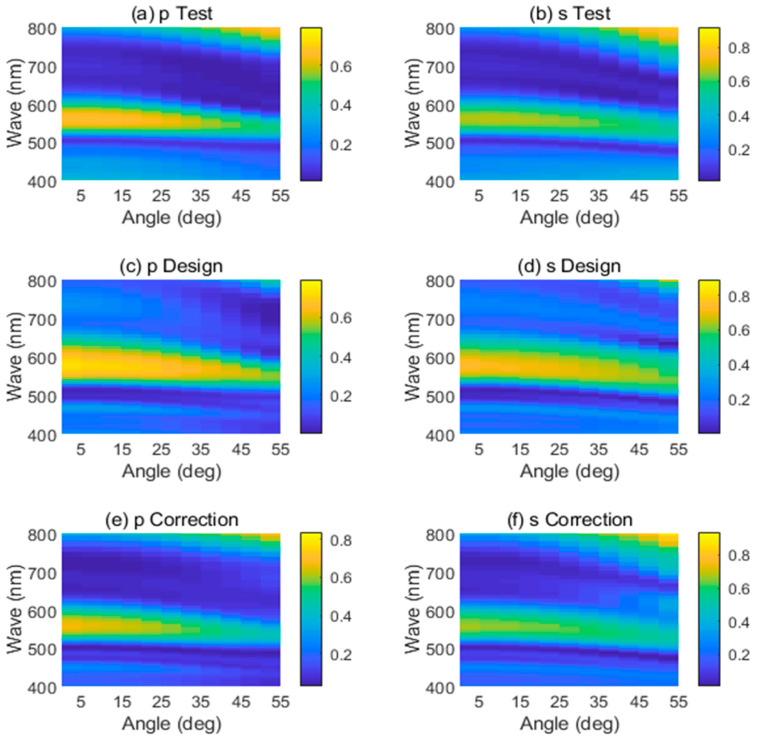
Reflectance spectra were measured in (**a**) p-polarization and (**b**) s-polarization. Reflectance spectra were simulated at (**c**) p-polarization and (**d**) s-polarization based on the designed film thickness values. Simulated reflectance spectra with actual film thickness values for (**e**) p-polarization and (**f**) s-polarization.

**Figure 4 materials-16-02316-f004:**
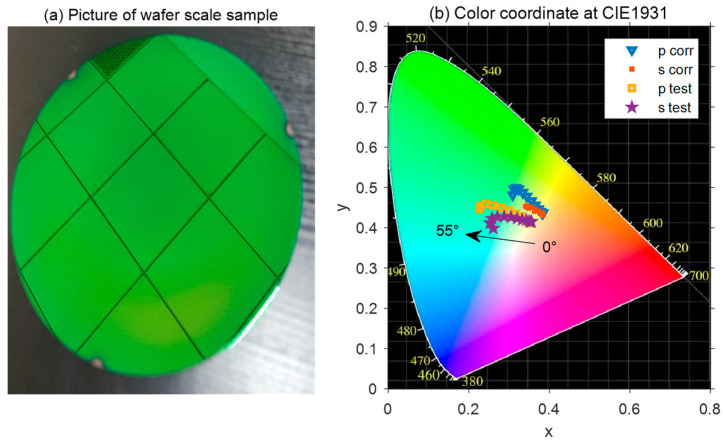
(**a**) Photograph of the green reflector at 8-inch wafer scale. (**b**) The CIE 1931 chromaticity diagram, showing the color coordinates in chromaticity diagram for the green reflector with the angle of incidence varied from 0° to 55°. The simulation with actual thickness and experimental test results are presented in the CIE 1931 chromaticity diagram, respectively.

**Table 1 materials-16-02316-t001:** SiN and SiO_2_ thin films deposition process parameters.

Materials	Parameters	Stable	Deposition	
			Stage 1	Stage 2
SiO_2_	Source RF (W)	0	300	300
	Bias RF (W)	0	0	150
	N_2_O (sccm)	40	40	40
	SiH_4_ (sccm)	20	20	20
SiN	Source RF (W)	0	300	
	Bias RF (W)	0	200	
	SiH_4_ (sccm)	55	55	
	N_2_ (sccm)	30	30	
	Ar (sccm)	200	200	

**Table 2 materials-16-02316-t002:** Film thickness values and percentage deviation of each layer under simulation and fabrication.

Layers	Materials	Simulation	Fabrication	Percentage Deviation
		Thickness (nm)		
1	SiO_2_	280	223	−20.4%
2	SiN	70	72	+2.9%
3	SiO_2_	118	110	−6.8%
4	SiN	140	146	+4.3%
5	SiO_2_	118	108	−8.5%
6	SiN	280	273	−2.5%
7	SiO_2_	118	114	−3.4%

Note: The sequence of each thin film from the surface of silicon substrate is symbolized by numbers.

**Table 3 materials-16-02316-t003:** The central wavelength and reflectance peak of the spectra in the p/s-polarization state at different angles.

Reflection Spectra	Polarization	$\lambda_{0}$ (nm)		Reflectance Peak	
		0°	55°	0°	55°
Measured spectra	p	563	533	66.9%	45.9%
	s	560	524	66.9%	49.6%
Simulated spectra of the designed structure	p	578	548	72.7%	52.6%
	s	578	534	72.7%	56.5%
Simulated spectra with actual thickness	p	562.5	534	66.7%	40.9%
	s	562.5	521	66.7%	48.1%

## Data Availability

The data presented in this study are available from the corresponding authors upon reasonable request.

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
