# Peer review of "Green Reflector with Predicted Chromatic Coordinates"

_materials, 2023, doi:10.3390/ma16062316_

Round 1

Reviewer 1 Report

This manuscript reported the Green Reflector with Predicted Chromatic Coordinates. This article is fixed with the scope of our journal. However, there are some problems for authors to improve their article before acceptance.

1. The capture of all figures should be revised. It is too long and same wording, must be shortened and more concise!!!

2. During the ICPCVD process, how long time for sputtering in each layer? It is should be provided in the Experimental section.

3. The composition of elements of each layer has to verify. Therefore, the EDX spectra of Figure 2b should be provided. What are P1-P4 in Figure 2a?

4. How about the hardness of each layer? and what is the interaction force between each layar? XPS should be investigating for verifying the chemical surface.

Reviewer 2 Report

Basically, the manuscript presents the design and construction of a well-executed monochromatic reflector, in a sufficiently demanding manner. My comments  about the work can be found below.

The concept, construction and technology are fundamentally similar to the color filter gescribed in reference 21, published in Sci Rep 2014. The authors should emphasize the differences more. The most important difference is the use of SiN compared to a:Si. Based on the present manuscript, the referee could not decide whether the choice of SiN was made because of the supposedly simpler technology or because of the smaller extension coefficient of the composite.

The authors deal with the Transfer Matrix Method and the  Finite-Difference Time-Domain algorithm in  separate chapter. Based on the TMM method, only the result shown in Fig 2a was calculated, but the conclusion of the manuscript refers only to the results of the FDTD calculation. Based on this, it seems unnecessary to discuss the TMM theory, since the simulated reflectance spectrum of the designed structure can also be derived from the TDTD theory.

The abbreviation SiN appears at the beginning of the manuscript without justification. It is important and must be emphasized that this is a silicon-rich, highly nonstoichiometric amorphous silicon nitride thin layer.

Reviewer 3 Report

The authors convincingly and clearly demonstrated a green reflector using a multilayer SiO2/SiN structure on silicon. A significant contribution to the results obtained was made by preliminary modeling of the specified optical parameters and subsequent comparison of the theoretical parameters of the proposed device with experimental ones . The work is well designed and can be published in a journal . However, the authors should make the following additions:

1. In the Introduction and Abstract, formulate the purpose of the work.

2. In conclusion, emphasize the novelty of the results obtained.

3. Use words in keywords, not abbreviations.

Reviewer 4 Report

The manuscript “Green Reflector with Predicted Chromatic Coordinates” deals with the design and experimental synthesis of multilayer structures with tunable optical properties, namely color reflectors. The results are interesting and clearly presented. There are however some points where clarifications or corrections are needed, as follows:

 Abstract: line 21, please rephrase “polarized light wave with an angle of incidence” with “polarized light wave with a variable angle of incidence”

Intoduction

Line 41 and 42, please avoid the use of term “researchers”.

Line 58 to 69, too much information is provided regarding the results. Please simplify this paragraph to convince the reader to continue, without revealing all the major findings from the beginning.

Methods and methodology

Lines 103-107, the designed structure is described. Please specify what was the desired “ideal” structure that was envisaged by this design (peak reflectivity, FWHM, maximum reflectance etc). Moreover, please specify how this specific structure was chosen, what are the advantages as compare with other structures.

 Experiment

Line 155: please explain the remark “to make sure a higher refractive index can be achieved.” In relation with the higher SiH4 flow

Line 157, please explain what does it mean that the gas ration follows the stoichiometry? If it refers to the stoichiometry f the thin films please specify how it was evaluated.

Line 160, line 169 ..etc. please use present tense instead of past tense : “is displayed” “is presented” etc

 Line 163-165, please rephrase, it is not clear what it means.

Lines 167, please define what “stable” means, what is the criterion?

 Line 217 and 231, please identify the “reflectance spectrum with actual film thickness of each layer obtained from the SEM image was 217 calculated by the TMM method” as being the  “correction” in the figure

 In figure 2 there is quite big difference between the initial design and the correction. This is most probably due, as stated by the authors, by the differences between the desired thicknesses and the actually obtained ones (table 2). It would be interesting to know if the authors attempted to replicate the design experimentally or this is the only one, and presumably the best matched with the design. From table 2 one might assume that the highest influence is given by the first layer, having ~20 % deviation. Fixing this deviation should partly fix the mismatch between design and experiment. Please evaluate this and detail it. How would the calculated reflectivity be with a 0% deviation on layer 1 and all the other deviations from table 2?

 Line 279, please specify what was the white balance setting on the camera. Incorrect white balance can modify the color appearance in the image

 Line 308, it is said that is critical to control the thickness. Please see my previous comment regarding the attempts to obtain other experimental multilayers and compare them.

Round 2

Reviewer 1 Report

Thank you for the satisfactory revision.

Author Response

Thanks for the comments.